# Assessing women's preferences towards tests that may reveal uncertain results from prenatal genomic testing: Development of attributes for a discrete choice experiment, using a mixed-methods design

Jennifer Hammond[1,2‡], Jasmijn E. Klapwijk[3‡], Sam Riedijk[3], Stina Lou[4], Kelly E. Ormond[5¤a], Ida Vogel[4], Lisa Hui[6,7,8], Emma-Jane Sziepe[9,10], James Buchanan[11,12], Charlotta Ingvoldstad-Malmgren[13,14,15], Maria Johansson Soller[13], Eleanor Harding[16], Melissa Hill[1,2‡], Celine Lewis[1‡¤b]*

1 North Thames Genomic Laboratory Hub, Great Ormond Street Hospital, London, United Kingdom, 2 Genetics and Genomic Medicine, UCL Great Ormond Street Institute of Child Health, London, United Kingdom, 3 Department of Clinical Genetics, Erasmus MC, Rotterdam, The Netherlands, 4 Center for Fetal Diagnostics, Department of Clinical Medicine, Aarhus University, Aarhus, Denmark, 5 Department of Genetics and Stanford Center for Biomedical Ethics, Stanford University School of Medicine, Stanford, CA, United States America, 6 Department of Obstetrics and Gynaecology, University of Melbourne, Parkville, Victoria, Australia, 7 Department of Perinatal Medicine, Mercy Hospital for Women, Heidelberg, Victoria, Australia, 8 Department of Obstetrics and Gynaecology, Northern Health, Epping, VIC, Australia, 9 Reproductive Epidemiology, Murdoch Children's Research Institute, Royal Children's Hospital, Parkville, Victoria, Australia, 10 Department of Paediatrics, Faculty of Medicine, Dentistry and Health Sciences, University of Melbourne, Melbourne, Australia, 11 Health Economics Research Centre, Nuffield Department of Population Health, University of Oxford, Oxford, England, United Kindom, 12 National Institute for Health Research Oxford Biomedical Research Centre, Oxford, England, United Kindom, 13 Department of Clinical Genetics, Karolinska Hospital and Karolinska Institutet, Stockholm, Sweden, 14 Center for Fetal Medicine, Karolinska University Hospital, Stockholm, Sweden, 15 Department of Molecular Medicine and Surgery, Karolinska Institutet, Stockholm, Sweden, 16 BSc Paediatrics and Child Health, The UCL Great Ormond Street Institute of Child Health, London, United Kingdom

¤a Current address: Health Ethics and Policy Lab, Department of Health Science and Technology, ETH Zurich, Zurich, Switzerland
¤b Current address: Population, Policy and Practice, UCL Great Ormond Street Institute of Child Health, London, United Kingdom
‡ JH and JEK share first authorship on this work. MH and CL are joint senior authors on this work.
* celine.lewis@ucl.ac.uk

**Data Availability Statement:** All relevant data are within the paper and its Supporting Information files.

**Funding:** CL received a Wellcome Trust Small Grant in Humanities and Social Science to conduct this work (grant number 211288/Z/18/Z). https://wellcome.org/grant-funding/schemes/small-grants-humanities-and-social-science JB received travel support from Illumina to attend conferences.

## Abstract

Prenatal DNA tests, such as chromosomal microarray analysis or exome sequencing, increase the likelihood of receiving a diagnosis when fetal structural anomalies are identified. However, some parents will receive uncertain results such as variants of uncertain significance and secondary findings. We aimed to develop a set of attributes and associated levels for a discrete-choice experiment (DCE) that will examine parents' preferences for tests that may reveal uncertain test results. A two phase mixed-methods approach was used to develop attributes for the DCE. In Phase 1, a "long list" of candidate attributes were identified via two approaches: 1) a systematic review of the literature around parental experiences of uncertainty following prenatal testing; 2) 16 semi-structured interviews with parents who had experienced uncertainty during pregnancy and 25 health professionals who return

The funders had no role in study design, data collection and analysis, decision to publish, or preparation of the manuscript.

**Competing interests:** The authors have read the journal's policy and have the following competing interests: JB received travel support from Illumina to attend conferences. There are no patents, products in development or marketed products associated with this research to declare. This does not alter our adherence to PLOS ONE policies on sharing data and materials.

uncertain prenatal results. In Phase 2, a quantitative scoring exercise with parents prioritised the candidate attributes. Clinically appropriate levels for each attribute were then developed. A final set of five attributes and levels were identified: likelihood of getting a result, reporting of variants of uncertain significance, reporting of secondary findings, time taken to receive results, and who tells you about your result. These attributes will be used in an international DCE study to investigate preferences and differences across countries. This research will inform best practice for professionals supporting parents to manage uncertainty in the prenatal setting.

## Introduction

Most women have routine ultrasound scans to be reassured about the health of their baby. In a small number of cases (around 4%), an unexpected structural anomaly will be detected [1]. In such cases, women can choose to have further investigative invasive procedures (e.g. amniocentesis or chorionic villus sampling (CVS)) with DNA testing conducted on the extracted sample. Chromosomal microarray analysis (CMA), which detects micro-deletions and -duplications in fetal DNA, is now routinely conducted as a first-line DNA test and has been shown to increase diagnostic yield over traditional karyotyping [2]. Prenatal exome sequencing (ES), which allows multiple genes to be screened simultaneously, is now entering research and clinical practice, and has been shown to increase diagnostic yield compared to CMA alone [3–5].

Prenatal testing following receipt of an abnormal ultrasound scan provides many clinical benefits, including the potential to provide a definitive diagnosis during pregnancy, which can provide useful information for pregnancy and delivery management [6, 7] and facilitate reproductive autonomy and psychological preparation [8]. However, a key concern is that the volume of information generated by these tests increases the incidence of uncertain results, including the identification of genomic variants of uncertain significance (VUS) or the detection of findings that are unrelated to the original reason for testing. Uncertain results can be difficult to deal with for parents who enter into prenatal testing looking for reassurance or definitive answers [9]. Previous research examining parents' experience of uncertainty following receipt of CMA or ES results has shown that whilst they are interested in receiving uncertain results, they are often surprised when they receive them and can experience shock, confusion, anxiety and decisional regret [10–14]. Understanding the preferences and priorities of parents for tests that may reveal uncertain results is important, and may help health professionals (HPs) working within genomics to identify the best way to support parents in such a scenario. For example, if the DCE reveals that one particular attribute is consistently more important than others, healthcare providers could ensure that care is taken not to focus on that one issue and to discuss a broad range of test features. A DCE may also reveal where differences of opinion arise amongst parents (e.g. whether older mothers or second-time mothers have different priorities and preferences regarding the return of uncertain prenatal test results than younger mothers), or even across countries, which may support varied approaches and guidelines to service delivery.

One method to examine people's values and preferences when making decisions is a discrete choice experiment (DCE). DCEs are a survey-based approach to elicit the preferences of participants by asking them to make trade-offs between different attributes (characteristics) of an intervention [15]. Participants are presented with a series of choices where at least two alternatives are specified in terms of their attributes, which can vary across a fixed number of levels. Participants are asked to complete these choice tasks, and regression analysis is used to develop a model of

choice behaviour [15]. DCE methods have been used widely within different areas of healthcare including genetics and genomic testing [16–21]. Their application in prenatal settings has commonly focused on comparing different methods of conducting prenatal tests [22, 23].

When developing DCEs, attributes should be selected that reflect the essential characteristics of the product or intervention, are considered important, are understandable and are mutually exclusive [24]. The number of attributes chosen should be a manageable number; most DCEs present between four to eight attributes [25]. Too many attributes increases the complexity of the task for respondents which may increase the chance of inconsistent responses across choice tasks or responders not considering all the attributes when making a decision [26]. Additionally, appropriate levels that are deemed "plausible, and capable of being traded" must be defined [25, 27]. Several methods can be applied to develop attributes and levels, including literature reviews, focus groups, interviews or consultations with key stakeholders, patient surveys, and expert reviews [25]. The importance of qualitative work when developing DCE attributes has been emphasised [24, 25, 28]. Notably, guidance on the conduct of DCEs has highlighted the lack of rigour in reporting attribute development [24, 25, 29, 30].

In this paper, we describe the use of both qualitative and quantitative methods to develop DCE attributes for an international comparison study that will examine patient preferences for receiving uncertain genomic test results in the prenatal setting.

## Materials and methods

A *clinical advisory group* (five HPs with expertise in prenatal genomics and fetal medicine from the UK, USA, Australia and Singapore) provided input into attribute development and assignment of clinically relevant levels. Ethical approval for this study was granted by the UK National Health Service Health Research Authority London–Riverside. REC reference: 18/LO/2120. Written consent was provided by those participants taking part in interviews conducted face-to-face; verbal consent (approved by the ethics committee) was provided and documented by the interviewer in telephone interviews where written consent could not be obtained.

A sequential mixed-methods approach across two phases was used to develop the attributes for the DCE (Fig 1). During **Phase 1**, we aimed to understand the different types of uncertainty that arise following CMA and/or ES. To do this we undertook a systematic review and conducted semi-structured interviews with parents and HPs. From this work, we developed a list of candidate attributes that were important to the study population and were capable of being traded. We also considered attributes used in existing DCE's within the field of genomics or prenatal testing as a means of cross-checking against our own list of attributes to identify gaps and inform attribute descriptions. **Phase 2** focused on reducing the candidate attribute list to those considered most important to parents using quantitative and qualitative methods, then determining the number of levels and their content.

### Phase 1: Attribute development

**Systematic review.**    Following methodological recommendations for the development of a DCE [24], we began by identifying potential attributes in the relevant published literature. We conducted a mixed-methods systematic literature review of women's views and experiences of uncertainty in pregnancy following CMA or ES. We aimed to understand the different sources of uncertainty that were encountered and how that uncertainty was managed in the clinical setting [31]. Studies were included if they were:

1. Investigating pregnant women and partners' experiences of uncertainty through the process of having CMA or ES;

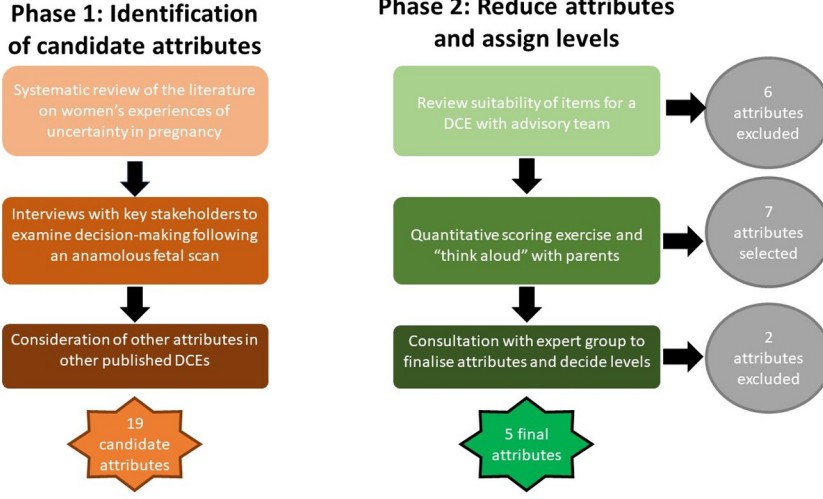

**Fig 1. Development phases.**

2. Using qualitative, quantitative, cross-sectional or mixed-methods research approaches;

3. Published in English in a peer-reviewed journal.

Studies were excluded if they were:

1. Investigating experiences of uncertainty not identified following CMA or ES, such as risk scores following Down syndrome screening, non-invasive prenatal testing or karyotyping;

2. Investigating parents' experiences following newborn or paediatric CMA and ES;

3. Examining views of uncertainty based on purely hypothetical scenarios;

4. A review, case report, abstract, editorial or commentary.

We searched three electronic databases (PubMed, PsycINFO and Embase) using relevant keywords (S1 Fig). The reference lists of eligible studies were searched, as well as other studies by JH. The initial search was conducted in October 2018. A further search was conducted in July 2019 and no additional papers were identified. The results of the identified studies were synthesised using the principles of thematic analysis [32] and meta-ethnography, which allows integration of findings across different study designs [33].

**Qualitative interviews with key stakeholders.** Semi-structured interviews were conducted with two groups of stakeholders:

1. HPs (clinical scientists, geneticists, genetic counsellors, fetal medicine consultants, obstetricians and paediatricians) from Australia, Denmark, the Netherlands, Singapore, Sweden and the UK working in prenatal testing with experience reporting or returning CMA and/or ES results.

2. Parents from the UK and the Netherlands who had experienced uncertainty in their pregnancy following an anomalous fetal scan where the implications for the baby were unclear (and not suspected to be Down syndrome).

Parent participants in the UK were recruited using the social media pages of the charity Antenatal Results and Choices (ARC) and through Great Ormond Street Hospital (GOSH) in London. In the Netherlands, parent participants were recruited via a Clinical Geneticist at

Erasmus Medical Centre in Rotterdam. Interviews were conducted with stakeholders from two different countries to ensure the chosen attributes would be widely relevant. Interviews with parents were conducted by CL and JH in the UK, and by JEK in the Netherlands. Interviews with HPs were conducted by the co-authors in their respective countries, with the exception of Singapore where interviews were conducted by a co-author from the UK (Australia—EJS, Denmark—SL, The Netherlands—JEK, Sweden–CI-M, UK—CL and EH, Singapore–CL). Interviews were conducted in the native (or national in the case of Singapore) language (other than Sweden where they were conducted in English), then translated into English by members of the research team (who are bilingual and work in two languages in their daily professional capacity). Full details of sampling, recruitment and data analysis are published elsewhere [13, 34].

*Topic guide*. Draft interview topic guides were developed by CL, JH and MH based on the findings of the systematic review and were revised with input from the wider research team (S2 and S3 Figs). Topic guides focused on what different types of uncertain results interviewees had come across, and how those results were managed. We also provided interviewees with a list of different types of uncertain results that had been identified from the systematic review (VUS, secondary findings, variants with reduced penetrance and variants with variable expression) and asked them whether these results should be fed back to parents.

*Data analysis*. Data were collected and analysed concurrently. Interviews were audio-recorded, transcribed verbatim and translated into English. Transcripts were coded and analysed using thematic analysis [32] using an abductive approach, which engages in a two-way dialogue between data and theory [35]. This approach was suitable for the qualitative analysis of this study, where we would be drawing together constructs from Han's taxonomy to explain and apply context and meaning to the data obtained [35]. The parent and HP interviews were analysed as two independent data sets. Data collection ceased when data saturation was reached, and no new themes or codes were emerging from the interview data. To ensure inter-researcher reliability, two researchers coded and categorised both datasets and the findings were discussed by all members of the research team.

**Integration of findings.** To produce an initial long list of potential attributes, the findings of the systematic review and stakeholder interviews were collated and compared. We focused on identifying attributes that reflected the **sources of uncertainty** that can be experienced following prenatal genomic testing that could be quantifiable as attributes with multiple levels in a DCE. To aid in understanding, we referred to a taxonomy of uncertainty by Han et al [36]. We also included attributes related to the **management** of uncertain prenatal test results because these were considered important by stakeholders in coming to a decision [37] and could mitigate against the impact of the uncertainty (where management was well done) or could enhance the sense of uncertainty (where management was poorly done).

**Consideration of attributes in other published DCEs.** Attribute development was undertaken using an inductive approach whereby our attributes were derived through research conducted with key stakeholders. To check whether we had missed any relevant attributes and to consider how others had framed similar concepts in previous research, we also reviewed other DCEs in the fields of prenatal testing, CMA and/or exome/genome sequencing (see S1 Fig for list of search term used). Attributes from these DCEs were considered alongside those identified in Phase 1 of our study as a form of cross-checking.

## Phase 2: Reducing the number of attributes and the development of levels

The long list of attributes was discussed with the research team, with the aim of removing any attributes that were 1) not quantifiable and therefore would not be feasible in the context of a DCE, 2) related to the condition being tested for rather than being an attribute of the test or its

**Table 1. Participant characteristics.**

| Parents (N = 16) | |
|---|---|
| **Gender** | |
| Female | 15 |
| Male | 1 |
| **Age** | |
| 20–30 | 6 |
| 31–40 | 10 |
| 41–50 | 2 |
| 51–60 | 0 |
| 61+ | 0 |
| **Self-Reported ethnicity** | |
| Caucasian | 6 |
| Asian/Asian British | 3 |
| Black/Black British | 0 |
| Other | 0 |
| Unreported | 7 |
| **Had invasive testing** | |
| Yes | 11 |
| No | 5 |
| **Had exome sequencing during pregnancy** | |
| Yes | 7 (only Dutch participants) |
| No | 9 |
| **Highest educational qualification** | |
| High School | 1 |
| Bachelor's degree | 15 |
| Professionals (N = 25) | |
| **Gender** | |
| Female | 21 |
| Male | 4 |
| **Age** | |
| 20–30 | 0 |
| 31–40 | 9 |
| 41–50 | 9 |
| 51–60 | 6 |
| 61+ | 1 |
| **Current role** | |
| Clinical scientist | 5 |
| Geneticist | 13 |
| Genetic Counsellor | 2 |
| Fetal Medicine Consultant | 1 |
| Obstetrician/Gynaecologist | 3 |
| Paediatrician | 1 |
| **Years in current role** | |
| 5–10 | 9 |
| 11–20 | 11 |
| 21–30 | 3 |
| >30 | 2 |

delivery i.e. the condition is not an essential characteristic of the intervention, 3) not relevant to clinical practice i.e. they could not be used to guide recommendations for delivering prenatal genomic tests and dealing with uncertainty. The refined list of attributes was then reviewed by:

1. a sub-group of the parent participants from the UK and the Netherlands who had taken part in the stakeholder interviews in Phase 1;

2. a patient advocate from the support group Antenatal Results and Choices (ARC); and

3. parents who had had a pregnancy in the previous three years (who were known to the authors), and had *not* experienced uncertainty linked to an anomalous fetal scan during that pregnancy. We did this to seek representative views from women with different experiences of pregnancy

We used a quantitative scoring exercise to rank the importance of each attribute. Similar quantitative approaches have been used in other DCEs to identify those attributes considered most important [38–40]. Each participant was presented with the list of attributes and asked to score the importance of each attribute on a scale of 1 (not important) to 5 (most important). As they were scoring each attribute, they were asked to verbalise ("think aloud") their decision-making process. This process was conducted either face-to-face or via telephone with one of the researchers (JH, CL or JEK), with qualitative and quantitative data captured on a score sheet.

The final step was to discuss the mean 'importance' scores with the research team and clinical advisory group to identify attributes that were the *most* relevant to uncertainty in a prenatal testing setting. For each of the final attributes, levels were chosen that represented a realistic range (as identified by the literature e.g. for diagnostic yield, or related to current practice e.g. for who returns results), over which DCE responders were expected to make trade-offs. Potential levels were discussed and agreed during a face-to-face meeting with our clinical advisory group.

## Results

### Phase 1: Attribute development

Our systematic review identified fourteen studies (ten qualitative, four quantitative) that met our inclusion criteria [14]. These studies were set in the USA, UK, Australia and the Netherlands, and captured the views of 914 participants (678 women, 236 partners). Interview participants included 16 parents from the UK (n = 9) and Netherlands (n = 7) who had experienced uncertainty following the detection of an undiagnosed fetal anomaly, 11 of whom had gone on to have invasive testing (Table 1) and 25 HPs (clinical scientists, consultants in clinical genetics, obstetricians and genetic counsellors) from the UK (n = 6), the Netherlands (n = 6), Denmark (n = 5), Singapore (n = 4) and Australia (n = 4) (Table 1).

Overall, 19 candidate attributes were identified from the systematic review and interviews (Table 2). The candidate attributes were categorised as either 'Sources of uncertainty' (i.e. the type of uncertainty) or 'Management of uncertainty' (i.e. how the uncertainty is managed inside and outside the clinic including service-related issues). Nine candidate attributes were regarded as 'Sources of uncertainty'; of these seven attributes were drawn from more than one dataset (HP interviews, patient interviews and systematic review) and two were drawn from the HP interviews only (Table 2). Ten candidate attributes were regarded as 'Management of uncertainty'. Of these, eight were drawn from both parent and HP datasets, one was drawn from the HP dataset, and one further attribute was drawn from the patient dataset.

**Comparison with attributes in other published DCEs.** We identified several previously conducted DCEs relevant to prenatal testing, CMA, exome and genome sequencing [16, 17, 19, 20, 22, 41–43]. The most commonly utilised attributes in these studies overlapped with

**Table 2. Candidate attributes and their sources.**

| Attribute | Illustrative quote | Attribute origin | | | Final Outcome |
|---|---|---|---|---|---|
| **Sources of uncertainty** | | SR[a] | HP[b] | PI[c] | |
| Uncertainty related to gene-disease correlations (genotype-phenotype correlations). | *"So a genotype-phenotype correlation is when I find something that will give a clinical picture that may not fit what you see on the ultrasound." HP interview—Dutch HP1* | ✓ | ✓ | | Excluded prior to scoring: Not considered quantifiable |
| Uncertainty about how a genetic anomaly with a well-known postnatal phenotype presents prenatally. | *"For a lot of disorders, we have no idea what it is prenatally, what the prenatal phenotype is because it is not published." HP interview Dutch HP5* | | ✓ | | Excluded prior to scoring: Not considered quantifiable |
| Pathogenicity and variants of uncertain significance | *"I'd want to know [this information] but then I'm thirsty for information. There's some people who I guess don't want to know but, for me, definitely." Parent interview—UK P7* | ✓ | ✓ | ✓ | Included in final attribute list |
| Penetrance (chance of having the phenotype) | *"Even though there's uncertainty, I think it's very different to be told there is, we have found something but not everybody gets it. I think that is. . .well in my mind, that's not too bad, but you haven't got that instant fear, oh god is it going to be me, it's like well OK it could be, that's fine. So I think that's a totally fine thing to say to somebody." Parent interview—UK P4* | ✓ | ✓ | ✓ | Excluded prior to scoring: Uncertainty related to the condition itself, and not the genetic test |
| Variable expressivity | *"For most participants, however, the lack of information about the likelihood and degree to which their fetus might be affected exacerbated concern. For example, one woman stated, "The part that freaked us out was definitely the spectrum to which they had associated my specific duplication. . .I don't feel like there was enough information." Systematic review quote from—Walser et al., 2016* | ✓ | ✓ | ✓ | Excluded prior to scoring: Uncertainty related to the condition itself, and not the genetic test |
| Likelihood of getting a result (Diagnostic yield) | *I: "Do you think it would still be worth doing it if the chances of picking up something with exome sequencing were really low?"* *R: "What do you mean really low? Because then I would argue that ten percent was low but we're doing CMA, so you still need to offer a test to those people. And if you stop offering CMA and you move onto whole genome sequencing for example, that's what you're offering". HP interview—UK HP1* | ✓ | ✓ | ✓ | Included in final attribute list |
| Secondary findings (these may be identified purposively or incidentally) | *"I think if it's a gene that could potentially lead to cancer or whatever, I mean, we could all be walking around with those, sometimes maybe it's best not to know. I personally would choose not to know that because I'm not going to terminate a child's life based on that, you know, so what good is it, you're going to just worry about it." Parent interview—UK P6* | | ✓ | ✓ | Included in final attribute list |
| Technical validity of test | *"It didn't perhaps dawn on us as to just how limited that test was, in that it was only looking for chromosomal abnormalities and that's not really what they were suspecting anyway." Parent interview—UK P1* | ✓ | ✓ | ✓ | Excluded prior to scoring: Not considered quantifiable |
| Possible incomplete result | *"We have had two cases I think where we've got one pathogenic variant in a recessive gene which fits the phenotype with no second hit, so that's also a difficult scenario where you'd go to MDT saying 'I've got one pathogenic variant and the gene doesn't fit with the phenotype, is it worth reporting?'". HP interview—UK HP6* | | ✓ | | Excluded prior to scoring: Not considered quantifiable |
| **Management of uncertainty** | | | | | |
| Who conducts pre-test counselling and delivers results | *"I think people with specialist knowledge of it for the specialist genome stuff, which would be people that work, like geneticists, like not just your general O & G [obstetrics and gynaecology] specialist." HP interview—Australian HP2* | ✓ | ✓ | | Included in final attribute list |
| Length of pre-test counselling | *"Right now, I try to get the pre-test down to approximately 45 minutes" HP interview—Danish HP4* | | ✓ | | Excluded following scoring |
| Who conducts post-test counselling | *"You want to be sure that you are dealing with someone who understands it and not someone uhh .. whose field of expertise it really isn't." Parent interview Dutch P5* | ✓ | ✓ | ✓ | Included in final attribute list |

*(Continued)*

**Table 2.** (Continued)

| Attribute | Illustrative quote | Attribute origin | | | Final Outcome |
|---|---|---|---|---|---|
| **Sources of uncertainty** | | SR[a] | HP[b] | PI[c] | |
| Length of post-test counselling | "it became obvious that if different parameters are selected for what results to return including variants of uncertain significance, genes of uncertain significance, or secondary findings, this will require longer more nuanced discussions." Systematic review—quote from Wou et al., 2018 | ✔ | ✔ | ✔ | Excluded following scoring |
| How results are delivered (face to face, phone etc) | The majority of participants (>18 participants) thought that a phone call was appropriate to disclose results. At least 2 participants expressed the need for time to process the information before reconvening to review the results. Systematic review—quote from Wou et al., 2018 | ✔ | ✔ | ✔ | Excluded following scoring |
| Communication style of person delivering counselling | "I think just that human touch, that human element that sort of, you know, common sense stuff which can easily be forgotten to address in all of this, really, really, really does make a big difference." Parent interview–UK P1 | ✔ | | ✔ | Excluded following scoring |
| Turnaround time for test results | "Gosh, well in an ideal world, they would be 24 hours! But I think a week maybe would be reasonable." Parent interview–UK P6 | ✔ | ✔ | ✔ | Included in final attribute list |
| Additional support available for patients | HP: "if they are overwhelmed with the result and they need to talk to mental wellness service, the service is readily available for them."<br><br>I: "So which service was that, sorry?"<br><br>HP: "Mental wellness service, like a psychologist." HP interview—Singapore HP4 | ✔ | ✔ | ✔ | Excluded following scoring |
| Who decides what results are fed back | "If it's a novel unknown then there's generally a discussion with other clinical scientists. If we are still unsure we might take it to a clinical geneticist or we might discuss it with some of the genomic curators." HP interview—Australian HP4 | | ✔ | ✔ | Excluded following scoring |
| Cost of test | "Found this a hard question, depends on the abnormality. If not considered severely debilitating, wouldn't want to pay that much (if at all) but if it was a potentially serious condition, then £2000". Parent interview UK P2 | | | ✔ | Excluded following scoring |

[a]Systematic review.

[b]Health professional interview data.

[c]Patient interview data.

those identified through the systematic review and interviews: accuracy of test, likelihood of condition developing, reporting of VUS, reporting of secondary findings, time taken to receive results and cost. Consequently, we used these previously published DCEs to conceptualise our candidate attributes, and their wording.

## Phase 2: Reducing the number of attributes and the development of levels

Following discussion with the research team, four attributes were removed because they were not considered to be quantifiable (genotype-phenotype correlations; how an anomaly with a well-known postnatal phenotype presents prenatally; technical validity of the test, and incomplete result). Two further attributes (penetrance and variable expressivity) were removed because the uncertainties related to the condition itself, rather than the genomic test (our DCE will ask participants to choose between two tests).

The remaining thirteen attributes were presented to stakeholders who represented parent views. In the UK, nine participants reviewed the attributes: three had experienced uncertainty

during their pregnancy and had undergone invasive testing, five were parents who had not experienced uncertainty during pregnancy, and one was a parent advocate. In the Netherlands, seven participants reviewed the attributes: four had experienced uncertainty during pregnancy and had undergone invasive testing, and three were parents who had not experienced uncertainty during pregnancy.

Six attributes were given a mean score of at least 4 (i.e. either important or very important). These were: Q4—length of time to get results (4.7), Q3a - secondary findings (of relevance to the baby) (4.6), Q11—communication style of HP delivering results (4.5), Q10—how results are delivered (4.4), Q1—diagnostic yield (4.4), and Q8—what type of HP delivers the results (4.0)(Table 3).

When discussing the **length of time to get a result**, participants felt that the *"test should be done properly"* but they *"wouldn't want to wait longer than 1 week"* for a result (UK P3), with one participant noting that *"24 hours would be ideal"* (UK P5). Regarding **secondary findings (of relevance to the baby)** parents stated that they would *"want to know everything for the baby"* (UK P5). However, one parent stated that wanting to know *"would depend on the severity. If severe, then definitely"* (Dutch P2). For **HP communication style** a *"compassionate communication style in times of stress"* (UK P4) was preferred. For **how results are delivered**, parents felt results should be *"relayed back in layman's terms, no jargon"* (UK P1). For **which HP should deliver results**, some parents noted a preference towards a genetics specialist returning results. However, one participant felt that who delivered results *"was not the most important, as long everything you need to know is fed back"* (UK P5), and another stated that the *"same person should be involved throughout the process"* (UK P2). How **results were delivered** could also depend on the severity of results: *"If a good result, then via the phone. But if it's a bad result, then personal contact"* (Dutch P4). The **diagnostic yield** of the test was frequently linked to the risk of miscarriage associated with invasive testing, with one participant stating that this factor is *"even more important"* when it is invasive (Dutch P4) and another saying that she *"wouldn't put [herself] through it, if there wasn't a good chance of getting an answer"* (UK P7).

Attributes that had lower mean importance scores included **secondary findings** (of relevance to parents) (3.2). One participant felt strongly that *"they wouldn't want to know everything"* (UK P2) when it came to secondary findings, and another echoed that they *"wouldn't want to worry unnecessarily"* about such findings (UK P3).

The **cost of the test** (1.8), including the role cost would play in determining whether one would choose to have a test or not, was also considered to be less important, regardless of which country the participant was from. Participants reported that, if required, they would be willing to pay for this test (ranging from £500 to £2000). However, for one participant, paying any price at all, could depend on the abnormality, which *"If not considered severely debilitating, [she] wouldn't want to pay that much, if at all. But if it was a potentially serious condition, then [she] would pay"* (UK P2). For others, their own financial circumstances could be a deciding factor.

When we compared the results of women who had experienced uncertainty with women who had not, five of the same six attributes had a mean score of at least 4. However, the attribute **what type of HP delivers the results** had a mean score of 3.5 amongst those who hadn't experienced uncertainty, possibly reflecting that the specialist who delivers the result is of less importance when that result is clear, easy to explain and no further investigations are required. When taking into consideration which country the women were from (UK or the Netherlands), the same six attributes had a mean score of at least 4. However, there were two attributes that had a mean score of at least 4 in the Netherlands but a mean score of less than 4 in the UK, namely notification about the **identification of VUS** (4.5 v 3.0) and **who decides which results are fed back** (4.0 v 3.3) (See Table 3).

The results of the scoring exercise were discussed with the research team. It was agreed that notification about the identification of VUS should be included in the DCE, despite only

**Table 3. Raw mean importance score for each attribute.**

| ATTRIBUTES<br>A pregnant couple go to the hospital to have their routine 20 week ultrasound scan. At the appointment, an abnormality is picked up on the scan which may indicate the baby has a genetic condition. The couple are offered further prenatal testing to see if a disease-causing gene change can be found which explains the baby's condition. If you were this couple…... | Average scores for UK parents/ parent advocate (experienced uncertainty) n = 4 | Average scores for Dutch parents (experienced uncertainty) n = 4 | Average scores for UK parents (who did not experience uncertainty) n = 5 | Average scores for Dutch parents (who did not experience uncertainty) n = 3 | Mean importance score for UK and Netherlands combined |
|---|---|---|---|---|---|
| 1. How important would it be for you to know *the likelihood that the test will find the disease-causing gene change (diagnostic yield)*? | 4.3 | 4.6 | 4.8 | 4 | **4.4** |
| 2. How important would it be for you *to be told about all gene changes that were found in the baby, even those where doctors can't be certain that they are disease-causing (variants of uncertain significance)*? | 3 | 4.5 | 3.2 | 3 | 3.4 |
| 3a. How important would it be for you to be told about the ability of the test to identify additional findings, unrelated to the original reason for testing, that might have health implications for you (secondary findings)? | 3.3 | 3.5 | 2.4 | 4.3 | 3.4 |
| 3b. How important would it be for you to be told about the ability of the test to identify additional findings, unrelated to the original reason for testing, that might have health implications for your baby (secondary findings)? | 4.3 | 4.5 | 4.6 | 5 | **4.7** |
| 4. How important would it be for you *how long it takes to get your test results*? | 4.8 | 4.5 | 4.6 | 5 | **4.7** |
| 5. How important would it be for you *who decides what results are fed back to you e.g. the couple, the health professional delivering the result, the clinical scientist that is writing the results report, professional guidelines*? | 3.3 | 3.3 | 2.2 | 4.5 | 3.3 |
| 6. How important would it be for you *what type of health professional conducts pre-test counselling e.g. whether it's a midwife, obstetrician, genetic counsellor, or other type of health professional with appropriate training*? | 3.5 | 3.3 | 3 | 3 | 3.2 |
| 7. How important would it be for you *how long the pre-test counselling appointment lasts*? | 3.3 | 2.8 | 3 | 2.7 | 3.0 |
| 8. How important would it be for you *which health professional conducts post-test counselling and delivers results e.g. whether it's a midwife, obstetrician, genetic counsellor, or other type of health professional with appropriate training*? | 4.8 | 4 | 3.4 | 3.6 | **4.0** |
| 9. How important would it be for you *how long the post-test counselling appointment to explain the test results lasts*? | 3.9 | 2.8 | 4 | 3.3 | 3.5 |
| 10. How important would it be for you *how results are delivered e.g. face-to-face appointment with the health professional, phone call, letter*? | 4.3 | 4.5 | 4.6 | 4 | **4.4** |

*(Continued)*

**Table 3.** (Continued)

| ATTRIBUTES<br>A pregnant couple go to the hospital to have their routine 20 week ultrasound scan. At the appointment, an abnormality is picked up on the scan which may indicate the baby has a genetic condition. The couple are offered further prenatal testing to see if a disease-causing gene change can be found which explains the baby's condition. If you were this couple…… | Average scores for UK parents/ parent advocate (experienced uncertainty) n = 4 | Average scores for Dutch parents (experienced uncertainty) n = 4 | Average scores for UK parents (who did not experience uncertainty) n = 5 | Average scores for Dutch parents (who did not experience uncertainty) n = 3 | Mean importance score for UK and Netherlands combined |
|---|---|---|---|---|---|
| 11. How important would it be for you *what the communication style of the health professional delivering counselling is e.g. whether they are empathetic, cold, clinical etc*? | 4.1 | 4.8 | 4.6 | 4.6 | **4.5** |
| 12 How important would it be for you *whether there is additional support available e.g. counselling, support groups*? | 3.9 | 3.3 | 3.4 | 4 | 3.7 |
| 13a. How important a factor would the cost of the test be for you when deciding whether or not to have the test? | 2.3 | 1 | 1.6 | 2 | 1.7 |
| 13b. If this test was *not* available through the NHS, what would you be prepared to pay to have the test? [range] | £100-£500 | €500-€5000 | £500-£1000 | €500-€5000 | £100-£5000 |

achieving a mean importance score of 3.3. This attribute captured a type of uncertainty that we were particularly interested in, as variability exists in terms of how these results are handled (as identified through the HP interviews), and for which it would be important to comment on in future recommendations. It was also agreed to exclude the attribute **who decides what results are fed back** due to its low mean importance score (3.2).

The seven remaining attributes were then discussed with the clinical advisory group. They agreed that the attribute **communication style of HPs delivering results** should be excluded as HPs should always adopt an empathic style when speaking with patients, therefore the value placed on this particular attribute would not inform a change to clinical management. They also felt that the attribute **how results are delivered** should be excluded as these policies are generally decided at a departmental or hospital level and all patients receiving an uncertain result should be seen in person irrespective of how that result is initially delivered. The clinical advisory team agreed that all the attributes selected satisfied the essential characteristics of a DCE attribute in that they reflected the characteristics of prenatal genomic tests and their management, were considered important, were understandable and were mutually exclusive.

For each of the five remaining attributes, two to four clinically feasible levels were chosen that were grounded in reality yet in some cases i.e. diagnostic yield, represented the higher and lower ends of what was realistic to 'force' participants to make decisions and trade-offs. For example, the levels set for diagnostic yield were 5%, 30% and 60% as these represented the upper and lower limits of what has been found to be clinically feasible [3]. The final set of attributes and levels are presented in Table 4.

## Discussion

Uncertainty is not uncommon in genomic medicine and new genomic technologies such as ES increase the potential for inconclusive test results as well as VUS and secondary findings

**Table 4. Final list of attributes and levels.**

| Attributes | Levels |
|---|---|
| Likelihood of getting a definitive result (diagnostic yield) | 5 out of 100 cases<br>30 out of 100 cases<br>60 out of 100 cases |
| Variants of uncertain significance | Reported<br>Not reported |
| Secondary findings (baby) | Reported<br>Not reported |
| Time taken to receive results | 1 week<br>2 weeks<br>4 weeks |
| Which HP returns and explains results | Maternity care provider<br>Genetics specialist |

[44]. One of the key challenges in researching and addressing uncertainty in this context is conceptualising what uncertainty looks like and what it means for the target population. This paper describes the development of attributes for a DCE that will examine parents' preferences for tests that may reveal uncertain test results. Applying a mixed-methods approach, we undertook a qualitative analysis of the existing literature and interviewed parents and HPs to aid the development of attributes, using quantitative methods to refine the number of candidate attributes. The final attributes and levels were then agreed upon by an expert clinical group.

The final list of attributes reflects multiple aspects of uncertainty in a prenatal setting and includes potential sources of uncertainty and issues linked to its management (Table 4). The inclusion of VUS and secondary findings is timely because reporting guidelines and practices around whether these should be returned differ in this area both between and within different countries, as highlighted by our recent review of guidelines in this area [45], and HP and parent views have been reported to differ [46]. The likelihood of getting a result is also topical given that diagnostic yield has been found to vary considerably depending on whether the fetus has isolated or multiple anomalies [3]. Regarding time taken to receive results, parents waiting for ES results following the identification of a fetal anomaly have found the period long and trying [7] and studies have shown that some HPs including genetic counsellors as well as maternity HPs are concerned about returning these types of results to their patients and desire further guidance in this area [47–49].

Our international DCE employing these attributes will yield important and timely insights into which uncertain results should be returned to pregnant couples, and which attributes are most pressing when parents make decisions about prenatal genomic tests. In particular, we will identify the most important attribute to parents when making decisions and the relative importance of this attribute compared to the other attributes; whether there is heterogeneity in preferences across countries with differing cultures and healthcare systems, and across participant types (e.g. whether older women or women who have experienced uncertainty in a previous pregnancy place greater emphasis on certain attributes than others); and what proportion of women would not opt for an invasive test following receipt of an abnormal fetal anomaly scan result.

An important strength in the development of the attributes presented in this paper was the use of both qualitative and quantitative methods to identify attributes specifically related to uncertainty. By first conducting a systematic review on parents' experiences of uncertainty in the prenatal setting, we were able to identify a longlist of candidate attributes. However, identifying attributes and their levels exclusively on the basis of a literature review may lead to the non-inclusion of some important attributes [24]. Accordingly, we extended our findings from

the systematic review by conducting qualitative interviews with both parents and HPs. Triangulating parent and HP views enhances the study's credibility [50]. Our use of interviews addresses recommendations to include qualitative work in developing DCE attributes [24, 25, 28]. Whilst the majority of the attributes identified in our systematic review were found in the qualitative interviews (13 out of the initial list of 19), we did indeed identify six attributes that were *not* identified in the review, including three attributes identified through the interviews with parents (which is notable given that the review focused on the experience of parents). This highlights the importance of conducting qualitative work in the development of DCEs.

Coast and Horrocks highlight that a 'tension' can exist between the purpose of qualitative work (in obtaining deep understanding of the phenomenon) and the "reductive aim" of describing key concepts in as few attributes as possible [25]. Whilst it is possible that the attributes themselves do not do justice to the "complexity of the individuals' preferences" [25], we aimed to mitigate this issue by asking parent interviewees to review the attributes that were developed (member checking). Furthermore, including stakeholders from countries with differing cultures and healthcare systems increases the potential generalizability of our findings. Finally, we ensured that the attributes (and associated levels) satisfied the essential characteristics of a DCE attribute by validating them with a clinical advisory team.

Our method has several limitations. All but two parents from the UK were recruited from a parent support group (ARC), and may have had particularly negative experiences during their pregnancy that led them to seeking support. In addition, the parent sample recruited through ARC was relatively homogenous, particularly in terms of education level and gender. This may have impacted which attributes were considered most important, with for example, those considered most important to women being included in the final set. Further research with partners of women who have experienced uncertainty following a fetal anomaly would therefore be valuable. Furthermore, given the attributes selected reflect those considered most important to women, this in turn could impact those topics chosen for discussion by health professionals during the counselling session. It is therefore important that health professionals ensure the views and concerns of men are also identified and addressed.

We included women who had not experienced uncertainty during their pregnancy as a comparator, however these women were recruited via a convenience sample of individuals known to the researchers. This may have limited the representativeness of the sample, although there were few differences between the women who did and did not experience uncertainty in terms of their attribute preferences. Additionally, the sample size for the quantitative work was relatively small. Although agreement between the scores of parents who experienced uncertainty and those that hadn't was good, future studies could consider undertaking the scoring of potential attributes with a broader sample. This would enable the application of more complex quantitative methods to select attributes for the DCE, such as Rasch analysis or factor analysis." Another limitation is that only Dutch parents had direct experience of ES, so may have different experiences regarding uncertainty in prenatal testing. A further potential limitation is that we did not consider the inclusion of attributes that were related to the condition being tested for rather than being an attribute of the test or its delivery. Tolerance of uncertainty may be linked to condition severity and could consequently impact on uptake of prenatal genomic testing. However, our DCE will investigate preferences for test attributes and delivery of results, and will not generate information on potential test uptake. Finally, it could be argued that views regarding uncertainty may not be generalisable across countries with differing healthcare systems. However, both UK and Dutch participants had similar views regarding the most important attributes, as did our international research team and clinical advisory group.

## Conclusions

We have described the development of attributes for a DCE assessing preferences towards receiving uncertain results from genomic testing. Using a mixed-methods approach, we have identified a set of five attributes for use in a DCE survey, with input from parents, HPs and experts in prenatal genomics. These have been used in a DCE survey that has been translated into multiple languages and recently used internationally to assess and compare tolerance for uncertainty in prenatal testing, the results of which are currently being analysed.

## Supporting information

**S1 Fig. Search terms used for review of other DCEs.**
(DOCX)

**S2 Fig. Parent topic guides.**
(DOCX)

**S3 Fig. Health professional topic guides.**
(DOCX)

## Acknowledgments

We thank the parents and professionals who participated in the study. We would also like to thank the clinical advisory group for their expert advice during the development of the attributes: Lyn S. Chitty, Eva Pajkrt, Mahesh Choolani, Louise Wilkins-Haug, Igna Van den Veyver, Catia Bilardo and Jane Halliday.

JB received travel support from Illumina to attend conferences. The other authors have no conflict of interest to report.

## Author Contributions

**Conceptualization:** Sam Riedijk, Stina Lou, Kelly E. Ormond, Ida Vogel, Lisa Hui, James Buchanan, Charlotta Ingvoldstad-Malmgren, Melissa Hill, Celine Lewis.

**Data curation:** Jasmijn E. Klapwijk, Sam Riedijk, Stina Lou, Emma-Jane Sziepe, Charlotta Ingvoldstad-Malmgren, Maria Johansson Soller, Eleanor Harding, Melissa Hill, Celine Lewis.

**Formal analysis:** Jennifer Hammond, Jasmijn E. Klapwijk, Emma-Jane Sziepe, Eleanor Harding, Melissa Hill, Celine Lewis.

**Funding acquisition:** Sam Riedijk, Stina Lou, Kelly E. Ormond, Ida Vogel, Lisa Hui, Charlotta Ingvoldstad-Malmgren, Melissa Hill, Celine Lewis.

**Investigation:** Jennifer Hammond, Emma-Jane Sziepe, James Buchanan, Melissa Hill, Celine Lewis.

**Methodology:** Jennifer Hammond, Sam Riedijk, Stina Lou, Kelly E. Ormond, Ida Vogel, Lisa Hui, James Buchanan, Charlotta Ingvoldstad-Malmgren, Melissa Hill, Celine Lewis.

**Project administration:** Jennifer Hammond, Melissa Hill, Celine Lewis.

**Supervision:** Sam Riedijk, James Buchanan, Melissa Hill.

**Validation:** Sam Riedijk, Stina Lou, Kelly E. Ormond, Ida Vogel, Lisa Hui, Charlotta Ingvoldstad-Malmgren, Maria Johansson Soller, Celine Lewis.

**Writing – original draft:** Jennifer Hammond, Celine Lewis.

**Writing – review & editing:** Jasmijn E. Klapwijk, Sam Riedijk, Stina Lou, Kelly E. Ormond, Ida Vogel, Lisa Hui, Emma-Jane Sziepe, James Buchanan, Charlotta Ingvoldstad-Malmgren, Maria Johansson Soller, Eleanor Harding, Melissa Hill.

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
