## [Decision Letter · Decision Letter 0]

18 Aug 2021

PONE-D-21-12941

Assessing women’s preferences towards tests that may reveal uncertain results from prenatal genomic testing: Development of attributes for a discrete choice experiment, using a mixed-methods design

PLOS ONE

Dear Dr. Lewis,

Thank you for submitting your manuscript to PLOS ONE. After careful consideration, we feel that it has merit but does not fully meet PLOS ONE’s publication criteria as it currently stands. Therefore, we invite you to submit a revised version of the manuscript that addresses the points raised during the review process.

As noted by Reviewers #1 and #2, the manuscript is well-written and relevant, and *PLOS ONE* would be a good outlet for this piece. Please attend to all reviewer comments described below, particularly with respect to both reviewers commenting on sample size and characteristics and Reviewer #2 requesting additional discussion and context on several components, such as methodology and future directions. Also, as noted by Reviewer #2, Table 4 was absent from the initial submission, so please include it in the revised manuscript. Finally, while not noted by either reviewer, when describing the results Table 3 in the main-text, it would be helpful to refer to which question #'s each construct referred to, so it facilitates interpretation by the reader; relatedly, with respect to Table 3 and the main text, there appear to be some discrepancies between the numbers in the table and those in the main text.

We look forward to receiving your revised manuscript.

Kind regards,

Andrew T Marshall, Ph.D.

Academic Editor

PLOS ONE

1. Please ensure that your manuscript meets PLOS ONE's style requirements, including those for file naming. The PLOS ONE style templates can be found at https://journals.plos.org/plosone/s/file?id=wjVg/PLOSOne_formatting_sample_main_body.pdf and https://journals.plos.org/plosone/s/file?id=ba62/PLOSOne_formatting_sample_title_authors_affiliations.pdf.

We will update your Data Availability statement to reflect the information you provide in your cover letter."

4. Please note that in order to use the direct billing option the corresponding author must be affiliated with the chosen institute. Please either amend your manuscript to change the affiliation or corresponding author, or email us at plosone@plos.org with a request to remove this option

Additional Editor Comments (if provided):

Reviewers' comments:

Reviewer's Responses to Questions

**Comments to the Author**

1. Is the manuscript technically sound, and do the data support the conclusions?

Reviewer #1: Yes

Reviewer #2: Yes

2. Has the statistical analysis been performed appropriately and rigorously? 

Reviewer #1: Yes

Reviewer #2: Yes

3. Have the authors made all data underlying the findings in their manuscript fully available?

Reviewer #1: Yes

Reviewer #2: Yes

4. Is the manuscript presented in an intelligible fashion and written in standard English?

Reviewer #1: Yes

Reviewer #2: Yes

5. Review Comments to the Author

Reviewer #1: This is an important and timely paper on the uncertainty around CMA/ ES results in a prenatal setting, and how healthcare professionals and patients and families might choose to deal with the information.

The methodology is appropriate, and the authors are well versed with this style of mixed-method evaluation of what might be otherwise subjective information that is difficult to analyse. Particularly commendable was the way data extracted from publications was combined with data obtained at interviews, in order to synthesise a logical approach to deploying the DCE methodology.

I am confident that the data presented here would form the basis of practice change in the clinical setting, but it is notable that the number of male participants amongst HP/ parents is very small. This may be a function of the subjects choosing to volunteer, or not, but this discrepancy in the ratios should be highlighted in the Discussion because it could have an impact on how the data from the study could be generalised into the counselling sessions in the clinic.

This paper addresses an important concern in daily clinical practice, that of VUS, and uncertainty, and in a setting not adequately investigated before: in the care of the pregnant woman with fetal abnormality.

Reviewer #2: I enjoyed reading this paper; it explained very well how they progressed from the qualitative work and quantitative work to the attributes for the DCE. The paper is well written. The paper would benefit from more reflection on the methods, and its contribution, as well as more detail on what the DCE method is and how it will be used. This latter point is important when deriving attributes. Some comments:

• Re the literature review and interviews, it would be good to know if they came up with different attributes. There is a big push currently to do extensive qualitative research in all DCEs. However, if extensive qualitative work has already been conducted, does doing more add to this? Reflections on this would give more depth to the paper. Did their qualitative work add anything to the literature review?

• Using quantitative methods to develop the attributes is not new; I am familiar with other studies that have done this. It would be good to put this study into context, referencing other studies. And perhaps some reflection on different quantitative methods to reduce a large set of attributes. I am aware of importance scores, factor analysis and Rasch analysis as methods to move from a longer set of attributes to a smaller set manageable within a DCE (see my point below re this).

• The disappointing part of this paper was the very small sample size for the quantitative work. Can the authors extend the sample size, or at least justify the small sample size.

• The paper is missing Table 4 – the final set of Attributes and Levels. I was keen to see this as the qualitative and quantitative work focused on attributes, and importance might depend on levels. I was also unclear how levels have been developed/defined from the qualitative work.

• Whilst I’m not a big fan of the BWS approach, a benefit of the Type 2 approach has been argued to be that importance is anchored on the levels of one attribute, rather than the attribute itself. This relates to the above point.

• I often ponder whether to publish the research developing attributes and levels as a standalone paper, or to include as part of the main DCE, often drawing on the Online Supplementary Information or a project Report to detail the study. My concern with this paper is that if you are not a DCE person it would not be clear why attributes have to be reduced to a small number, and indeed, what can be done with a DCE over and above finding out what is important.

• Related to the above point, there may be reasons for including a cost attribute beyond the importance score i.e. the research wants to look at value in monetary terms. The paper would benefit from some discussion of the aim of the DCE.

• The key contribution of the DCE approach is that it enables trade-offs to be made – I’m a bit unclear what the trade-offs are here. This relates to the analysis plan, how will the DCE data be used.

• The paper states: ‘This research will inform best practice for professionals supporting parents to manage.’ I am not sure how it will achieve this, but more detail of the attributes, levels and analysis plan would inform.

6. PLOS authors have the option to publish the peer review history of their article (what does this mean?). If published, this will include your full peer review and any attached files.

Reviewer #1: No

Reviewer #2: No

---

## [Author Response · Author response to Decision Letter 0]

28 Sep 2021

We very much thank the reviewers for their comments. These have been very helpful in improving the quality of this manuscript. 

Reviewer #1: This is an important and timely paper on the uncertainty around CMA/ ES results in a prenatal setting, and how healthcare professionals and patients and families might choose to deal with the information.

The methodology is appropriate, and the authors are well versed with this style of mixed-method evaluation of what might be otherwise subjective information that is difficult to analyse. Particularly commendable was the way data extracted from publications was combined with data obtained at interviews, in order to synthesise a logical approach to deploying the DCE methodology.

We thank the reviewer for their positive feedback. 

I am confident that the data presented here would form the basis of practice change in the clinical setting, but it is notable that the number of male participants amongst HP/ parents is very small. This may be a function of the subjects choosing to volunteer, or not, but this discrepancy in the ratios should be highlighted in the Discussion because it could have an impact on how the data from the study could be generalised into the counselling sessions in the clinic.

Thank you for this point. We recruited parents who had experienced a fetal anomaly through the support group ARC, and their membership tends to be, understandably, very female dominated. In addition, the majority of health professionals were female and this may possibly be because genetics is a relatively female-dominated profession; our recruitment rate was relatively good (79%) but it is not clear whether those who declined were male. 

We have addressed these issues in the Limitations, as the low number of male participants is a potential limitation of the study:

“In addition, the parent sample recruited through ARC was relatively homogenous, particularly in terms of education level and gender. This may have impacted which attributes were considered most important, with for example, those considered most important to women being included in the final set. Further research with men who have experienced uncertainty following a fetal anomaly would therefore be valuable. Furthermore, given the attributes selected reflect those considered most important to women, this in turn could impact those topics chosen for discussion by health professionals during the counselling session. It is therefore important that health professionals ensure the views and concerns of men are also identified and addressed." 

This paper addresses an important concern in daily clinical practice, that of VUS, and uncertainty, and in a setting not adequately investigated before: in the care of the pregnant woman with fetal abnormality.

We agree this is an important area to focus on given the increase in the number of uncertain results that will occur following more detailed testing. 

Reviewer #2: I enjoyed reading this paper; it explained very well how they progressed from the qualitative work and quantitative work to the attributes for the DCE. The paper is well written. 

We thank the reviewer for this feedback.

The paper would benefit from more reflection on the methods, and its contribution, as well as more detail on what the DCE method is and how it will be used. This latter point is important when deriving attributes. Some comments:

• Re the literature review and interviews, it would be good to know if they came up with different attributes. There is a big push currently to do extensive qualitative research in all DCEs. However, if extensive qualitative work has already been conducted, does doing more add to this? Reflections on this would give more depth to the paper. Did their qualitative work add anything to the literature review?

Thank you for raising this point. Much of this is detailed in Table 2 where we show where each attribute was identified. In the majority of cases (13 out of the initial list of 19), attributes were identified across both the interviews (with patients, HPs or both) and the systematic review. However, there were 6 attributes identified from our qualitative interviews that were not found in the initial systematic review we conducted. In three cases, attributes were identified through interviews with parents that were not identified in the systematic review that focused on parents’ experiences. This is a notable finding as it points to the importance of interviews in the development of DCEs. We have added this reflection in the Discussion section of the paper. 

We have added the following text to address this point in the Discussion:

“Whilst the majority of the attributes identified in our systematic review were found in the qualitative interviews (13 out of the initial list of 19), we did indeed identify six attributes that were not identified in the review, including three attributes identified through the interviews with parents (which is notable given that the review focused on the experience of parents). This highlights the importance of conducting qualitative work in the development of DCEs.” 

• Using quantitative methods to develop the attributes is not new; I am familiar with other studies that have done this. It would be good to put this study into context, referencing other studies. And perhaps some reflection on different quantitative methods to reduce a large set of attributes. I am aware of importance scores, factor analysis and Rasch analysis as methods to move from a longer set of attributes to a smaller set manageable within a DCE (see my point below re this).

We have referenced three studies that have used similar quantitative approaches to reduce the initial list of candidate attributes. As the reviewer notes, more complex quantitative methods exist to develop attributes, however, these alternative approaches require a larger sample size for the underlying qualitative work than we had available in this study. We have added a note in the limitations section of our Discussion to address this comment – please see the following response.

"Additionally, the sample size for the quantitative work was relatively small. Although agreement between the scores of parents who experienced uncertainty and those that hadn’t was good, future studies could consider undertaking the scoring of potential attributes with a broader sample. This would enable the application of more complex quantitative methods to select attributes for the DCE, such as Rasch analysis or factor analysis." 

• The disappointing part of this paper was the very small sample size for the quantitative work. Can the authors extend the sample size, or at least justify the small sample size.

Our small sample size for the quantitative part of the study reflects our recruitment approach where we went back to the 16 parents who were part of the interview study and unfortunately not all agreed. We did, however, include a patient representative from ARC who has the insight of many years speaking to parents about uncertainty in pregnancy and then matched the sample with a comparison group. We appreciate this is a small number for a quantitative study. However, when we compared the mean scores of parents who experienced uncertainty with those that hadn’t, five of the same six attributes scored at least 4 or above. This indicated very little difference in the views of women towards which were the most important attributes.

We have now addressed this point in the Limitations as per the above. 

• The paper is missing Table 4 – the final set of Attributes and Levels. I was keen to see this as the qualitative and quantitative work focused on attributes, and importance might depend on levels. I was also unclear how levels have been developed/defined from the qualitative work.

Apologies – these have now been added to the paper. The levels were decided by our clinical advisory team (rather than through the qualitative work) and were either yes/no e.g. secondary findings reported, or were levels that were grounded in reality. We have provided some further detail in the paper in the Methods and Results:

“For each of the final attributes, levels were chosen that represented a realistic range (as identified by the literature e.g. for diagnostic yield, or related to current practice e.g. for who returns results)…”

“For example, the levels set for diagnostic yield were 5%, 30% and 60% as these represented the upper and lower limits of what has been found to be clinically feasible [42].”

• Whilst I’m not a big fan of the BWS approach, a benefit of the Type 2 approach has been argued to be that importance is anchored on the levels of one attribute, rather than the attribute itself. This relates to the above point.

As noted above, further details about the development of the levels has now been provided in the paper. 

• I often ponder whether to publish the research developing attributes and levels as a standalone paper, or to include as part of the main DCE, often drawing on the Online Supplementary Information or a project Report to detail the study. My concern with this paper is that if you are not a DCE person it would not be clear why attributes have to be reduced to a small number, and indeed, what can be done with a DCE over and above finding out what is important.

We also pondered whether to publish this as a standalone paper or as supplementary material inside the main paper. In the end we decided to publish this as a standalone paper a) because we wanted to contribute to the literature in this area, particularly given there are relatively few methodological papers reporting the development of attributes for DCE’s, and b) because it seemed a shame to put so much important detail into the supplementary material, which would not be as visible. 

We have added some further detail about the number of attributes in the Introduction to address the point related to the reason why the number of attributes has to be reduced to a manageable number:

“The number of attributes chosen should be a manageable number; most DCEs present between four to eight attributes .; Too many attributes increases the complexity of the task for respondents which may increase the chance of inconsistent responses across choice tasks or responders not considering all the attributes when making a decision [26].” 

In terms of what can be done with a DCE beyond finding out what is important to parents, we have also added this additional description of the value of DCE results:

“For example, if the DCE reveals that one particular attribute is consistently more important than others, healthcare providers could ensure that care is taken not to focus on that one issue and to discuss a broad range of test features. They may also reveal where differences of opinion arise amongst patients (e.g. whether older mothers have different priorities and preferences regarding the return of uncertain prenatal test results than younger mothers), or even across countries, which may support varied approaches and guidelines to service delivery.

• Related to the above point, there may be reasons for including a cost attribute beyond the importance score i.e. the research wants to look at value in monetary terms. The paper would benefit from some discussion of the aim of the DCE.

As noted on page 5, the overall aim of our DCE was to quantify the preferences of patients in multiple countries for receiving uncertain genomic test results in the prenatal setting. Including a cost attribute in healthcare-related DCEs is always challenging, because in many settings patients are unaware of the cost of providing healthcare, so do not have well-formed preferences for a cost attribute. For example, in taxpayer- or social insurance-funded health systems, patients do not pay directly for healthcare. Approaches exist to frame cost attributes for DCEs in these settings (e.g. asking respondents to imagine that their taxes will increase by £x to cover the cost of providing a new healthcare intervention), but this may not entirely resolve the issue. Complicating this issue further is the fact that this DCE survey will be conducted across multiple countries, all of which differ in how healthcare is funded. In this scenario it is challenging to interpret the results of a DCE including a cost attribute. For these reasons, in addition to the low importance score, we decided not to include a cost attribute in this DCE.

• The key contribution of the DCE approach is that it enables trade-offs to be made – I’m a bit unclear what the trade-offs are here. This relates to the analysis plan, how will the DCE data be used.

We have now provided further detail about the analysis plan in the Discussion:

“In particular, we will look at: which is the most important attribute to parents when making decisions and its relative importance in comparison to the other attributes; whether there is heterogeneity in preferences across countries with differing cultures and healthcare systems, and across participant types (e.g. whether older women or women who have experienced uncertainty in a previous pregnancy place greater emphasis on certain attributes than others); and what proportion of women would not opt for an invasive test following receipt of an abnormal fetal anomaly scan result.”

• The paper states: ‘This research will inform best practice for professionals supporting parents to manage.’ I am not sure how it will achieve this, but more detail of the attributes, levels and analysis plan would inform.

The final list of attributes and levels (Table 4) has now been added into the manuscript and some further detail about the development of the levels has been provided at the end of the Methods section. In terms of the analysis plan, we have added additional information, as detailed above. 

Editors comment

When describing the results Table 3 in the main-text, it would be helpful to refer to which question #'s each construct referred to, so it facilitates interpretation by the reader; relatedly, with respect to Table 3 and the main text, there appear to be some discrepancies between the numbers in the table and those in the main text.

We have now added the question numbers in the text and have also amended the discrepancies. Thank you for pointing this error out.

---

## [Decision Letter · Decision Letter 1]

14 Dec 2021

Assessing women’s preferences towards tests that may reveal uncertain results from prenatal genomic testing: Development of attributes for a discrete choice experiment, using a mixed-methods design

PONE-D-21-12941R1

Dear Dr. Lewis,

We’re pleased to inform you that your manuscript has been judged scientifically suitable for publication and will be formally accepted for publication once it meets all outstanding technical requirements.

Kind regards,

Andrew T. Marshall, Ph.D.

Academic Editor

PLOS ONE

Additional Editor Comments (optional):

Both reviewers agree that all comments have been addressed, so there are no further revisions.

Reviewers' comments:

Reviewer's Responses to Questions

**Comments to the Author**

1. If the authors have adequately addressed your comments raised in a previous round of review and you feel that this manuscript is now acceptable for publication, you may indicate that here to bypass the “Comments to the Author” section, enter your conflict of interest statement in the “Confidential to Editor” section, and submit your "Accept" recommendation.

Reviewer #1: All comments have been addressed

Reviewer #2: All comments have been addressed

2. Is the manuscript technically sound, and do the data support the conclusions?

Reviewer #1: Yes

Reviewer #2: (No Response)

3. Has the statistical analysis been performed appropriately and rigorously? 

Reviewer #1: Yes

Reviewer #2: (No Response)

4. Have the authors made all data underlying the findings in their manuscript fully available?

Reviewer #1: Yes

Reviewer #2: (No Response)

5. Is the manuscript presented in an intelligible fashion and written in standard English?

Reviewer #1: Yes

Reviewer #2: (No Response)

6. Review Comments to the Author

Reviewer #1: (No Response)

Reviewer #2: (No Response)

7. PLOS authors have the option to publish the peer review history of their article (what does this mean?). If published, this will include your full peer review and any attached files.

Reviewer #1: No

Reviewer #2: No

---

## [Editor Report · Acceptance letter]

20 Jan 2022

PONE-D-21-12941R1 

*Assessing women’s preferences towards tests that may reveal uncertain results from prenatal genomic testing: Development of attributes for a discrete choice experiment, using a mixed-methods design*

Dear Dr. Lewis:

I'm pleased to inform you that your manuscript has been deemed suitable for publication in PLOS ONE. Congratulations! Your manuscript is now with our production department. 

Kind regards, 

on behalf of

Dr. Andrew T. Marshall 

Academic Editor

PLOS ONE